# A New Approach to Modeling Ultrashort Channel Ballistic Nanowire GAA MOSFETs

**DOI:** 10.3390/nano12193401

**Published:** 2022-09-28

**Authors:** He Cheng, Zhijia Yang, Chao Zhang, Chuang Xie, Tiefeng Liu, Jian Wang, Zhipeng Zhang

**Affiliations:** 1Key Laboratory of Networked Control Systems, Chinese Academy of Sciences, Shenyang 110016, China; 2Shenyang Institute of Automation, Chinese Academy of Sciences, Shenyang 110016, China; 3Institutes for Robotics and Intelligent Manufacturing, Chinese Academy of Sciences, Shenyang 110169, China

**Keywords:** ballistic transport, compact model, sub-7 nm nanowire GAA MOSFET, smoothing function method

## Abstract

We propose a numerical compact model for describing the drain current in ballistic mode by using an expression to represent the transmission coefficients for all operating regions. This model is based on our previous study of an analytic compact model for the subthreshold region in which the DIBL and source-to-drain tunneling effects were both taken into account. This paper introduces an approach to establishing the smoothing function for expressing the critical parameters in the model’s overall operating regions. The resulting compact model was tested in a TCAD NEGF simulation, demonstrating good consistency.

## 1. Introduction

As electronic device dimensions shrink, nanoscale devices with multigate structures are considered to improve immunity to short-channel effects (SCEs), such as drain-induced barrier lowering (DIBL) and threshold voltage roll-off [1,2,3,4,5,6,7,8,9]. In recent years, several novel structures of devices [10,11,12,13,14,15,16,17], such as nanowire (NW) and nanosheet (NSH) metal-oxide semiconductor field-effect transistors (MOSFETs), have attracted more attention and become potential structures that most likely led to downscaling into the sub-7 nm regime. Then, because of the reduction in scattering events in the short-channel nanowire GAA MOSFETs, carriers traveling through the channel without scattering increase ballistic transport [18,19,20,21,22]. Thus, the properties of such devices in which the quantum transport of carriers should be taken into account were intensively investigated and modeled [23,24,25,26,27,28,29].

Moreover, since the DIBL effect cannot be neglected in an ultrashort channel gate-all-around (GAA) MOSFET, it is more meaningful to develop a compact model incorporating ballistic transport, quantum confinement, and quantum tunneling effects. Then, the ballistic transport and quantum mechanism in ultrashort channel nanowire GAA MOSFETs were widely recognized as significant factors in determining drain current characteristics by using a numerical calculation method such as Schrödinger–Poisson and nonequilibrium Green’s function (NEGF) simulations [30,31,32,33]. The results of the current characteristics from the TCAD simulation and the experiment both demonstrate higher $I_{\mathrm{DS}}$ and better subthreshold swing (SS) characteristics in a GAA MOSFET than those in a Fin field-effect transistor (FinFET) in a wide range of temperature. They also indicate that GAA MOSFETs can provide more robust solutions to next-CMOS devices beyond FinFETs and low-supply-voltage applications [29,34,35]. However, the computations of the simulations could be time-consuming, even for estimating the characteristics of one single device. A compact model of nanowire GAA MOSFETs would be more efficient in describing the drain-current characteristics for a reduction in computational time when taking the ballistic transport and quantum effects into account.

Several compact models evaluating GAA-MOSFET properties were reported [10,36,37,38,39,40,41,42,43,44,45,46]. Some models focused on the expressions of the drain-current characteristic with a DIBL effect, incorporating the transport modes on the basis of the drift-diffusion [39,44] and ballistic theories [41]. Although the DIBL effect is mentioned in some of those models, the quantum confinement [39,45] and the source-to-drain tunneling effects [10,41,42] also need to be considered in the models. Consequently, the drain current would be represented by combining tunneling and thermionic ballistic transport expressions in the subthreshold and inversion regions [47]. However, a fully analytic compact model that can be introduced to SPICE-class simulators using Verilog-A language to implement a circuit-level simulation is expected to be mathematically developed to predict the behavior of a nanowire GAA MOSFET as a final purpose. According to our previous work, the potential distribution of a sub-band energy level and transmission coefficient of the source-to-drain tunneling can be expressed and determined approximately by using one parabolic function and the Wentzel–Kramers–Brillouin (WKB) approximation, respectively. Although source-to-drain tunneling was incorporated in the ballistic device with an ultrashort channel according to our previous study [47], it is increasingly necessary to simplify the calculation process in a compact model when evaluating the drain current by only using the Landauer formula in all operating regions.

In this paper, we propose one numerical compact model of the nanowire GAA MOSFET with an ultrashort channel to describe the ballistic drain current by introducing an approach of the smoothing function to determine the significant parameters through connecting the expressions of the subthreshold and inversion regions mentioned in our previous study [24]. Hence, we improved expressing the unknown parameter, the barrier top locations, and the transmission coefficients for all operating regions using the smoothing functions mentioned above. Then, the drain current for the ultrashort channel GAA MOSFET is evaluated from the Landauer formula, which is in line with NEGF simulation results. The rest of this paper is organized as follows. We introduce the model structure, definition of parameters, and the ballistic current model in Section 2. Section 3 illustrates how we use the smoothing functions to approximately express the unknown parameter, the barrier top location, and the transmission coefficients being valid for all operating regions. Lastly, the calculation results and discussion are in Section 4.

## 2. Device Structure and Model Formulas

### 2.1. Model Structure

Figure 1a,b show the schematic structures of the modeled nanowire GAA MOSFETs in the channel and transverse directions, respectively. In this structural model, we consider the source and drain electrodes as the ideal reservoirs of electrons since they were produced with highly doped *n*-type silicon that would provide sufficient electrons into the intrinsic silicon channel surrounded by ${\mathrm{SiO}}_{2}$ gate oxide and metal gate with a circular cross-section. Moreover, the backscattering from the source and drain electrodes was neglected. Channel length, radius, and gate oxide thickness are denoted as $L_{G}$, *R*, and $T_{\mathrm{OX}}$, respectively. Correspondingly, the cylindrical coordinate with length, radius, and angular components along with *z*, *r*, and φ directions, as shown in Figure 1b, is used in the following formulations. However, on account of the perfect angular symmetry in our model, it was sufficient to represent the electrostatic potential distribution only along the channel and radius directions without considering the φ component.

### 2.2. Potential Profile

The profiles of the conduction band edge and the higher sub-band energy levels along the *z* and *r* axes at r=0 and $z=z_{\mathrm{MAX}}$ are shown in Figure 2a,b, respectively, where it was assumed that the short-channel effects to the channel along the transverse direction in this model were consistent. Then, as shown in Figure 2a, $E_{\mathrm{total}}$ is the total energy of an electron, and the maximal value of the sub-band energy $E_{n_{\varphi },n_{r},\mathrm{MAX}}$ within the channel was located at $z_{\mathrm{MAX}}$, where $n_{\varphi }$ and $n_{r}$ are the angular and radial quantum numbers, respectively. The energy value around the source edge within the same sub-band is denoted as $E_{n_{\varphi },n_{r},\mathrm{MIN}}$. Furthermore, $E_{F,S}$ and $E_{F,D}$ represent the Fermi energies at the source and drain, respectively, while the voltage across, expressing the difference of the electrostatic potential between the source and drain in Fermi level $\left(-E_{F,S}/e+E_{F,D}/e\right)$, is $V_{\mathrm{DS}}$. The electrostatic potentials in the transverse direction at the center and surface of the cylindrical channel are defined as $w_{0}$ and $w_{S}$, respectively, and the conduction band edges at the same locations are given by $-ew_{0}$ and $-ew_{S}$, respectively, as illustrated in Figure 2b. Correspondingly, the difference between $w_{0}$ and $w_{S}$ was described using an unknown parameter, $\Delta U_{G}$. Moreover, the higher sub-band energy levels measured from $E_{F,S}$ are $E_{n_{\varphi },n_{r}}$. According to our previous study, $E_{n_{\varphi },n_{r}}$ can be written in terms of $\Delta U_{G}$ along the *z* axis as follows [6,22]:(1)$$E_{n_{\varphi },n_{r}}\left(z\right)=\frac{\hbar^{2}}{2{m_{r}}^{2}}{\left(\frac{R_{n_{\varphi }}^{n_{r}}}{R}\right)}^{2}-eV_{\mathrm{GS}}^{*}+F\cdot \Delta U_{G}\left(z\right),$$
(2)$$F=e\left(H_{n_{\varphi },n_{r}}+\frac{\alpha }{R^{2}}-1\right),$$
where *ℏ*, $m_{r}$, and $R_{n_{\varphi }}^{n_{r}}$ are the Planck’s constant, the confinement electron effective mass in *r* axis, and the constant described as the $n_{r}$-th zero point of the first-kind Bessel function with order $n_{\varphi }$, respectively, $V_{\mathrm{GS}}$ is the gate-source voltage, $\varphi_{\mathrm{GC}}$ refers to the work function difference between the gate and the channel material, and $w_{\mathrm{FB}}$ is the conduction band edge at the flat-band condition corresponding to the relationship of $V_{\mathrm{GS}}^{*}=V_{\mathrm{GS}}-\varphi_{\mathrm{GC}}+w_{\mathrm{FB}}$. Moreover, $H_{n_{\varphi },n_{r}}$ is the perturbation matrix constant that was calculated in our previous work [24], α denotes a geometric constant defined by $R^{2}\left(1/2+4\pi \varepsilon_{\mathrm{CH}}/C_{\mathrm{OX}}\right)$, $\varepsilon_{\mathrm{CH}}$ is the dielectric constant of the channel, and $C_{\mathrm{OX}}$ is the gate oxide capacitance per unit length. $\Delta U_{G}$ along the *z* axis is given by
(3)$$\Delta U_{G}\left(z\right)=-N\cdot A\exp\left(\gamma \cdot z\right)-N\cdot B\exp\left(-\gamma \cdot z\right),$$
where *A* and *B* are the coefficients related to the voltage between the gate/drain and the source, respectively; γ is the scaling coefficient depending on α mentioned above that can be given by
(4)$$\gamma =\sqrt{\frac{4}{\alpha }}=\frac{2}{R\sqrt{\frac{1}{2}+\frac{4\pi \varepsilon_{\mathrm{CH}}}{C_{\mathrm{OX}}}}}.$$

Accordingly, the above coefficients are given by:(5)$$A=\left(V_{\mathrm{bi}}-V_{\mathrm{GS}}^{*}\right)\left[1-\exp\left(-\gamma \cdot L_{G}\right)\right]+V_{\mathrm{DS}},$$
(6)$$B=\left(V_{\mathrm{bi}}-V_{\mathrm{GS}}^{*}\right)\left[\exp\left(\gamma \cdot L_{G}\right)-1\right]-V_{\mathrm{DS}},$$
(7)$$N=\frac{128\gamma^{2}\cdot R^{2}}{\left(8-\gamma^{2}\cdot R^{2}\right)\left(64+\gamma^{4}\cdot R^{4}\right)}\cdot \frac{1}{\exp\left(\gamma \cdot L_{G}\right)-\exp\left(-\gamma \cdot L_{G}\right)},$$
where $V_{\mathrm{bi}}$ is the junction built-in potential between the source and channel that represents the electrostatic potential energy level of a sub-band at the source measured from $E_{F,S}/\left(-e\right)$ [6]. In addition, the corresponding position to the barrier top of the sub-band energy level along the channel can be derived with:(8)$$z_{\mathrm{MAX}}=\frac{1}{2\gamma }\ln\left(\frac{B}{A}\right).$$

### 2.3. Landauer Formula

Considering that the device channel length was shorter than 7 nm, electrons traveling through the device channel comprised the ones whose energies were higher or lower than the sub-band energy level at the barrier top. Thereby, the Landauer formula could determine the drain current contributed by the electrons in the ballistic transport between the source and drain with both the thermionic and tunneling modes. Then, the ballistic current between the drain and source in the channel can be written as follows [48]:(9)$$\begin{array}{c} I_{\mathrm{DS}}=\frac{e}{\pi \hbar }\sum_{n_{\varphi },n_{r}}\sum_{n_{\nu }}\int_{0}^{\infty }dE_{z}\cdot T_{n_{\varphi },n_{r}}\left(E_{z}\right)\left[f\left(E_{F,S},E_{\mathrm{total}}\right)-f\left(E_{F,D},E_{\mathrm{total}}\right)\right], \end{array}$$
(10)$$E_{\mathrm{total}}=E_{n_{\varphi },n_{r}}\left(z\right)+E_{z},$$
(11)$$f\left(E_{F},E\right)=\frac{1}{1+\exp\left(\frac{E-E_{F}}{k_{B}T}\right)},$$
where $n_{v}$ is the valley index of the energy band, $T_{n_{\varphi },n_{r}}\left(E_{z}\right)$ is the transmission coefficient of electrons tunneling through the source-to-drain barrier in the sub-band with quantum numbers of $n_{\varphi }$ and $n_{r}$ at a given $E_{z}$, which is the electron kinetic energy in the *z* direction, and $f(E_{F},E)$ refers to the Fermi–Dirac distribution. Here, $T_{n_{\varphi },n_{r}}\left(E_{z}\right)$ denotes the transmission probability derived from the WKB approximation:(12)$$T_{n_{\varphi },n_{r}}\left(E_{z}\right)=\left\{\begin{array}{cc} 1, & E_{z}>E_{n_{\varphi },n_{r}}\left(z_{\mathrm{MAX}}\right)\\ \exp\left\{-\frac{2}{\hbar }\int_{z_{1}}^{z_{2}}dz\cdot \sqrt{2m_{C}\left[E_{n_{\varphi },n_{r}}\left(z\right)-E_{z}-E_{n_{\varphi },n_{r}}\left(0\right)\right]}\right\}, & E_{z}<E_{n_{\varphi },n_{r}}\left(z_{\mathrm{MAX}}\right) \end{array}\right.$$
where $m_{C}^{*}$ is the electron effective mass along with the channel direction, and $z_{1}$ and $z_{2}$ are defined as the classical turning points at which $E_{z}$ is equal to the potential barrier energy.

## 3. Calculation Method

In this study, electrons carrying energy higher than the maximal potential barrier were transmitted from the source to the drain by thermionic emission. In comparison, electrons with lower energy can traverse the channel only by quantum tunneling through the potential barrier between the source and drain. However, it was assumed that electrons transporting from the drain could be neglected when calculating both the thermionic ballistic and the source-to-drain tunneling current calculations under the condition of high drain bias as $eV_{\mathrm{DS}}\gg k_{B}T$, regardless of the electrons traveling over and through the potential energy barrier. Moreover, it was assumed that the ballistic current consists of electrons only excited to the lowest sub-band when $n_{\varphi }=0$ and $n_{r}=1$ as the quantum numbers for all operating regions since no electrons are excited to the second or higher sub-band energy levels in the nanowire GAA MOSFET. Then, the carrier quantum reflection was considered to be negligible on the barrier in the channel. Nevertheless, in the channel of purely ballistic transport, the device behavior model could be simplified to describe the electrons’ transmission over and through the source-to-drain potential energy barrier in one consistent expression.

### 3.1. Parabolic Energy Level Approximation

Since WKB approximation was applied to calculate the transmission probability for the source-to-drain tunneling, it had to be evaluated by numerically integrating the classical turning points from Equation (Equation 12). Therefore, it could render the model analytically calculable by assuming that the sub-band energy profile along the device channel to be a parabolic function, which is valid in a short-channel device [47]. Then, when the channel length is sufficiently short, the lowest sub-band energy level along the *z* direction $E_{0,1}\left(z\right)$ could be assumed to be approximately described by a parabolic function of *z*, denoted as $E_{0,1,b}\left(z\right)$ [47]. This leads to the result below:(13)$$E_{0,1,b}\left(z\right)=C-D\cdot {\left(z-z_{\mathrm{MAX}}\right)}^{2},$$
(14)$$C=E_{0,1}\left(z_{\mathrm{MAX}}\right),$$
(15)$$D=\frac{E_{0,1}\left(z_{\mathrm{MAX}}\right)-E_{0,1}\left(0\right)}{{z_{\mathrm{MAX}}}^{2}},$$
where $E_{0,1}\left(z_{\mathrm{MAX}}\right)$ and $E_{0,1}\left(0\right)$ are given by Equation (Equation 1) at the positions when $z=z_{\mathrm{MAX}}$ and z=0, respectively, as follows:(16)$$E_{0,1}\left(z_{\mathrm{MAX}}\right)=\frac{\hbar^{2}}{2{m_{r}}^{2}}{\left(\frac{R_{0}^{1}}{R}\right)}^{2}-eV_{\mathrm{GS}}^{*}+F\cdot \Delta U_{G}\left(z_{\mathrm{MAX}}\right),$$
(17)$$E_{0,1}\left(0\right)=\frac{\hbar^{2}}{2{m_{r}}^{2}}{\left(\frac{R_{0}^{1}}{R}\right)}^{2}-eV_{\mathrm{GS}}^{*}+F\cdot \Delta U_{G}\left(0\right).$$

Then, by substituting $z=z_{\mathrm{MAX}}$ and z=0 into Equation (Equation 3), $\Delta U_{G}\left(z_{\mathrm{MAX}}\right)$ and $\Delta U_{G}\left(0\right)$ can be derived in terms of *A* and *B*, related to the bias conditions:(18)$$\Delta U_{G}\left(z_{\mathrm{MAX}}\right)=-2\cdot N\sqrt{A\cdot B},$$
(19)$$\Delta U_{G}\left(0\right)=-N\cdot \left(A+B\right).$$

### 3.2. Unknown Parameter for All Operating Regions

This study introduces and implements a mathematical approach for establishing the smoothing function by connecting two kinds of expressions in their different regions. In addition, this method requires two different expressions of an arbitrary function and a constant in the expressions’ respective ranges of values. Compared with the previous work on the unknown parameter at the top of the potential barrier in all operating regions [6], an alternative approach to represent $\Delta U_{G}\left(z_{\mathrm{MAX}}\right)$ was achieved. Implementing the smoothing function to represent $\Delta U_{G}\left(z_{\mathrm{MAX}}\right)$ in Equation (Equation 18), it was set as zero for a large gate voltage bias, while its expression was valid in the subthreshold region incorporating the DIBL effect. It could thus be rewritten as $\Delta U_{G}^{\mathrm{DIBL}}$ for all operating regions with the following formula:(20)$$\Delta U_{G}^{\mathrm{DIBL}}=2\cdot N\sqrt{\frac{1}{a_{1}}\cdot \ln\left[1+\exp\left(a_{1}\cdot A\cdot B\right)\right]},$$
where $a_{1}$ is a fitting constant to be adjusted for the best accuracy; this value was fixed to be $a_{1}=0.5$ for wide devices and bias conditions. To obtain a single valid expression for all operating regions, we blended $\Delta U_{G}^{\mathrm{DIBL}}$ with $\Delta U_{G}^{\left(1\right)}$ by following the same assumption as that in [24] as follows:(21)$$\Delta U_{G}^{\mathrm{all}}=\Delta U_{G}^{\mathrm{DIBL}}+\Delta U_{G}^{\left(1\right)}.$$

As a result, by substituting Equation (Equation 21) into Equation (Equation 1), we could rewrite the value of the lowest sub-band energy level at the barrier top $E_{0,1}\left(\Delta U_{G}^{\mathrm{all}}\right)$ for all operating regions as follows:(22)$$E_{0,1}\left(\Delta U_{G}^{\mathrm{all}}\right)=\frac{\hbar^{2}}{2{m_{r}}^{2}}{\left(\frac{R_{0}^{1}}{R}\right)}^{2}-eV_{\mathrm{GS}}^{*}+F\cdot \Delta U_{G}^{\mathrm{all}}.$$

### 3.3. Potential Barrier Top for All Operating Regions

Furthermore, from analyzing Equation (Equation 8), the expression of $z_{\mathrm{MAX}}$ was valid only in the subthreshold region and took meaningless values in the inversion region. According to the same inference in Section 3.2 to describe the smoothing function for a whole range of values, it could be assumed that $z_{\mathrm{MAX}}$, representing the distance between the locations of the source edge and the potential barrier top, decreased to nearly zero with the ever-increasing gate voltage. Thus, two expressions were needed to be given by Equation (Equation 8) and zero in the subthreshold and inversion regions, resectively. Then, the barrier top position in all operating regions $z_{\mathrm{MAX}}^{\mathrm{all}}$ was approximately evaluated as follows:(23)$$z_{\mathrm{MAX}}^{\mathrm{all}}=\frac{1}{2\gamma }\cdot \ln\left\{1+\frac{1}{a_{2}}\cdot \ln\left[1+\exp\left(a_{2}\cdot \frac{B}{A}\right)\right]\right\}.$$
where $a_{2}$ is a fitting parameter without any physical meaning; this value was fixed to be $a_{2}=0.95$ for wide devices and bias conditions to achieve reasonable accuracy.

### 3.4. Transmission Coefficient for All Operating Regions

According to Equation (Equation 12), the tunneling transmission coefficient through a sub-band energy barrier could be expressed in two conditions corresponding to a given $E_{z}$. Then, representing the sub-band energy level profile as a parabolic function, the tunneling transmission coefficient $T_{0,1}$ applying the WKB approximation to substitute Equation (Equation 13) into Equation (Equation 12) was determined as follows:(24)$$T_{0,1}\left(E_{z}\right)=\exp\left[-\frac{\pi }{\hbar }\sqrt{\frac{2m_{C}}{C_{0}}}\left(C_{0}-E_{z}\right)\cdot z_{\mathrm{MAX}}^{\mathrm{all}}\right].$$
(25)$$\begin{array}{cc} C_{0} & =E_{0,1}\left(\Delta U_{G}^{\mathrm{all}}\right)-E_{0,1}\left(0\right), \end{array}$$
(26)$$\begin{array}{cc} \; & =F\cdot \left[\Delta U_{G}^{\mathrm{all}}-\Delta U_{G}\left(0\right)\right]. \end{array}$$

On the other hand, the transmission coefficient of the energy sub-band here was unity and $T_{0,1}$ corresponding to the electrons with the total energies of $E_{\mathrm{total}}>E_{0,1,\mathrm{MAX}}$ and $E_{\mathrm{total}}<E_{0,1,\mathrm{MAX}}$, respectively. Here, $T_{0,1}$ varied between 0 and 1. Considering that the smoothing function requires the expressions given by $T_{0,1}$ and one in their ranges of values taking with $E_{z}+E_{n_{\varphi },n_{r}}\left(0\right)<E_{0,1,\mathrm{MAX}}$ and $E_{z}+E_{n_{\varphi },n_{r}}\left(0\right)>E_{0,1,\mathrm{MAX}}$, the transmission coefficient for all operating regions could then be represented as follows:(27)$$T_{0,1}^{\mathrm{all}}\left(E_{z}\right)\approx \exp\left\{-\frac{1}{a_{3}}\ln\left[1+\exp\left(a_{3}\cdot \frac{\pi }{\hbar }\sqrt{\frac{2m_{C}}{C_{0}}}\left(C_{0}-E_{z}\right)\cdot z_{\mathrm{MAX}}^{\mathrm{all}}\right)\right]\right\},$$
where $a_{3}$ denotes the fitting parameter, and the value was set to be $a_{3}=$ 1.5 to obtain a good agreement for all operating regions in the next section.

### 3.5. Drain Current for All Operating Regions

As illustrated in Figure 3, the transmission coefficient of the lowest sub-band here was unity and $T_{0,1}\left(E\right)$, corresponding to the electrons with energies higher or lower than the potential barrier top. For a high drain bias, the second term on the right-hand side of Equation (Equation 9) including $V_{\mathrm{DS}}$ could be neglected. Therefore, only the electrons injected from the source to drain could contribute to the ballistic drain current in this model. Since we could represent $\Delta U_{G}^{\mathrm{all}}$ and $T_{0,1}^{\mathrm{all}}$ in all operating regions using Equations (Equation 21) and (Equation 23), respectively, we demonstrate the drain current in all operating regions $I_{\mathrm{DS}}^{\mathrm{all}}$ according to our previous work. Then, the formula of the drain current is as follows:(28)$$I_{\mathrm{DS}}^{\mathrm{all}}\approx \frac{e\cdot g_{n_{\nu }}}{\pi \hbar }\int_{0}^{\infty }dE_{z}\cdot \frac{T_{0,1}^{\mathrm{all}}\left(E_{z}\right)}{1+\exp\left[\frac{E_{0,1}\left(0\right)+E_{z}}{k_{B}T}\right]}.$$

## 4. Results and Discussion

This section compares the results from our compact model and those from a TCAD simulation using SILVACO ATLAS [49]. Numerical computations were performed on the basis of the nonequilibrium Green’s function (NEGF) formalism under the effective mass approximation coupled with Poisson’s equation. Moreover, Wolfram Mathematica software was used to present the compact model. The cylindrical coordinate was used for quasi-three-dimensional calculation, no scattering was considered, and only the lowest sub-band was taken into account. The supply voltage was set to be 0.5 V following IRDS [50], and the $V_{\mathrm{bi}}$ value was extracted from the NEGF simulation.

As illustrated in Figure 4, the lowest energy level profiles between the NEGF simulation and our compact model using Equation (Equation 13) were compared. Three lines correspond to the results with $V_{\mathrm{GS}}=$ 0, 0.2, and 0.5 V from the top. The channel length used in this compact model was empirically expanded into the source and the drain regions by 0.5 and 1 nm throughout the comparisons between the calculations of numerical NEGF simulation and the compact model considering Equation (Equation 13) for good agreement, respectively, referring to [6].

Figure 5 shows the calculation results of $\Delta U_{G}^{\mathrm{all}}$ using a solid black line corresponding to Equation (Equation 21) in determining the unknown parameter at the barrier top for all operating regions, denoted as $\Delta U_{G}^{\mathrm{all}}$. The results from $\Delta U_{G}^{\mathrm{DIBL}}$ and $\Delta U_{G}^{\left(1\right)}$, corresponding to $\Delta U_{G}$ in the subthreshold and inversion regions, are also shown as the red and green dashed lines, respectively. When the DIBL effect was neglected, $\Delta U_{G}^{\left(1\right)}$ in the subthreshold region was 0. On the other hand, $\Delta U_{G}^{\mathrm{DIBL}}$ took a negative value in the subthreshold region, while it became 0 in the inversion region deriving from Equation (Equation 20). When r= 2.5 nm and $L_{G}=$ 4 nm, $\Delta U_{G}^{\mathrm{DIBL}}$, which should have been 0 in our mentioned assumption, took a negative value, even in the inversion region. Although this case could be fixed by resetting $a_{1}$ as 2 or a greater number, an inappropriate value of $a_{1}$ affected the accuracy of the calculations for $E_{0,1,\mathrm{MAX}}-V_{\mathrm{GS}}$, $T_{0,1}-E_{\mathrm{total}}$ and $I_{\mathrm{DS}}-V_{\mathrm{GS}}$ characteristics, as shown in the following discussion.

Figure 6 demonstrates comparisons of the dependencies for $z_{\mathrm{MAX}}$-$V_{\mathrm{GS}}$, corresponding to the potential barrier top positions in the device channel at different gate voltages resulting from the NEGF simulation and our compact model. The dashed lines, solid curves, and open circles show the calculations from the compact model by using Equations (Equation 8) and (Equation 23), and the NEGF simulation, respectively. The value of $z_{\mathrm{MAX}}$ calculated from (Equation 8) was valid only in the subthreshold region, and took an impossible value in the inversion region. Thereby, the necessity of deriving Equation (Equation 23) from (Equation 8), describing $z_{\mathrm{MAX}}$ as a non-negative value, speaks for itself. Moreover, when $V_{\mathrm{GS}}$ exceeded 0.5 V, the compact model could not capture the results of the barrier top positions from the NEGF simulation. This limited the valid range of the operating gate bias voltage in the compact model.

Figure 7 illustrates the gate voltage dependence of the lowest sub-band energy level at the barrier top $E_{0,1,\mathrm{MAX}}$ using Equation (Equation 22). The comparisons between the NEGF simulation and our compact model with the lowest sub-band are demonstrated as open circles and solid lines, respectively, maintaining good consistency.

Figure 8 shows comparisons of dependencies for $T_{0,1}$-$E_{\mathrm{total}}$, corresponding to electrons having the total energy of $E_{\mathrm{total}}$, between our numerical compact model using Equation (Equation 27) and NEGF simulator calculations. The open circles and solid lines relate to the calculations from the compact model and the NEGF simulation, respectively, demonstrating reasonable agreement between these results at each gate voltage level, from 0 to 0.4 V in 0.1 V steps.

Figure 9 illustrates the drain current versus gate voltage characteristics calculated at various sets of the channel lengths and wire radius by considering Equation (Equation 28) in all operating regions. The open circle and solid line represent the comparison results of the compact model and NEGF simulation calculation, respectively. As shown in this figure, our compact model could capture the drain current increase caused by the source-to-drain tunneling in the subthreshold region while maintaining acceptable accuracy in on-current characteristics. This model was not valid when the sub-band energy level at the barrier top was lower than that at the source, which determined the maximal value of the gate voltage bias to sustain the validity of the model by solving $E_{n_{\varphi },n_{r}}\left(z_{\mathrm{MAX}}\right)=E_{n_{\varphi },n_{r}}\left(0\right)$ using Equations (Equation 16) and (Equation 17). In our computations, these gate bias voltages were larger than the given supply voltage. Nevertheless, in order to obtain reasonable agreement for all operating regions, the diameter and length values of the channel needed to meet a ratio of less than unity ($2R/L_{G}$ < 1) in this study through their comparisons between the NEGF simulator and our compact model. Furthermore, the tunneling current should also be considered in the inversion region.

## 5. Conclusions

This paper presented a new approach to express the numerical compact model of the drain current in the ultrashort channel nanowire GAA MOSFET using the Landauer formula. Assuming that a parabolic function could approximately determine the potential distribution along the ultrashort channel, the WKB approximation was capable of evaluating the transmission coefficients for source-to-drain tunneling. In addition, it could approximately evaluate the position of the potential barrier top and the transmission coefficient for all operating regions by using smoothing functions. Lastly, the drain current characteristics could be calculated for all operating regions. This numerical compact model was tested with a NEGF simulation, demonstrating reasonable agreement in all operating regions. Although the compact model contained several empirical parameters, they were fixed throughout the comparisons. However, a compact model with fully analytic expressions is expected to be developed to carry out a circuit simulation in a future study.

## Figures and Tables

**Figure 1 nanomaterials-12-03401-f001:**
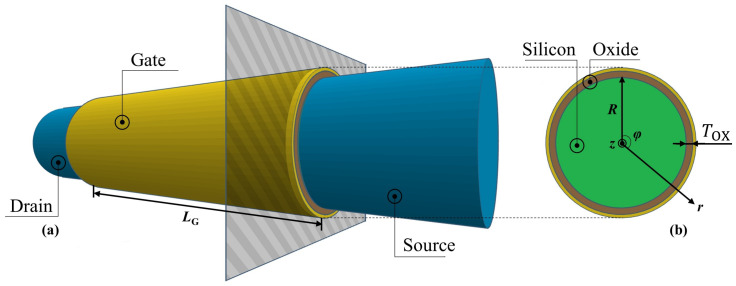
Schematic structure (**a**) in the channel direction *z* and (**b**) in the transverse direction *r* of the nanowire GAA MOSFET model.

**Figure 2 nanomaterials-12-03401-f002:**
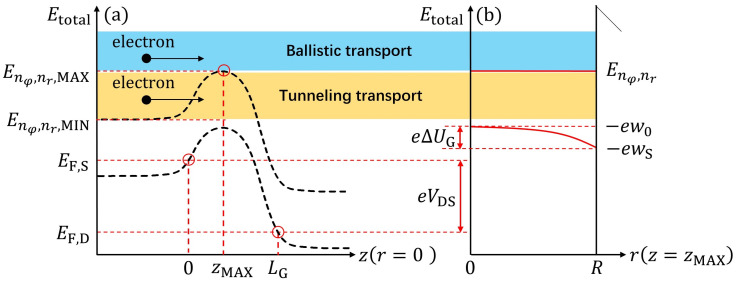
Rough sketch of the potential energy profiles along the channel and the transverse directions, and mechanisms governing the carrier transport in ballistic tunneling and thermionic modes. (**a**) Representation of energy-level distribution along the *z* direction at the channel center (r=0). (**b**) Schematics of confinement potential energy distribution along the *r* direction at the barrier top ($z=z_{\mathrm{MAX}}$) in the cross-section. The elementary charge is denoted by *e*.

**Figure 3 nanomaterials-12-03401-f003:**
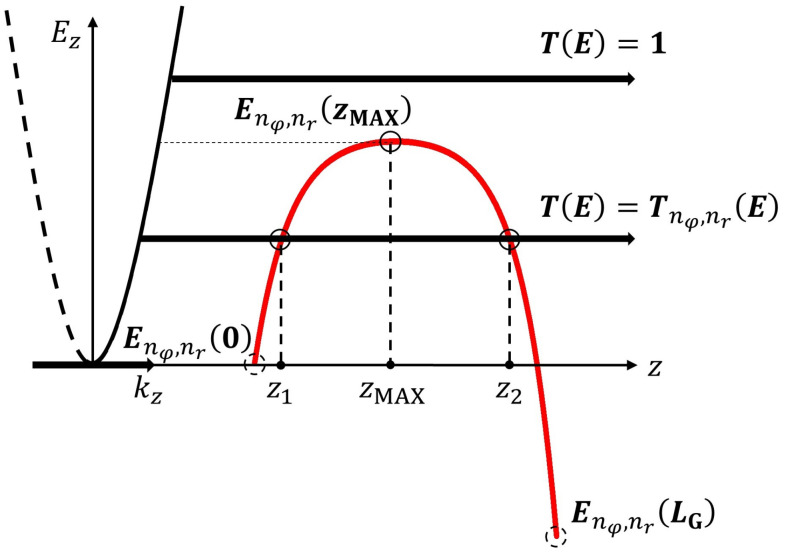
Schematic representation of electrons transported between the source and drain, corresponding to their total energy. Illustrations of the quantum tunneling and thermionic ballistic transport mechanism of electron transport across the potential barrier from the source to the drain along the channel. Sub-band energy profile $E_{n_{\varphi },n_{r}}\left(z\right)$ obtained from Equation (Equation 1) is represented. Turning points ($z_{1}$ and $z_{2}$) and $z_{\mathrm{MAX}}$ are shown in literal values. Here, $k_{z}$ is the electron wave vector component in the *z* direction corresponding to the parabolic dispersion relation for $E_{z}$. Schematic representation for the electrons transporting between the source and drain according to the total energy of electrons. Several possible cases for electron transport are demonstrated.

**Figure 4 nanomaterials-12-03401-f004:**
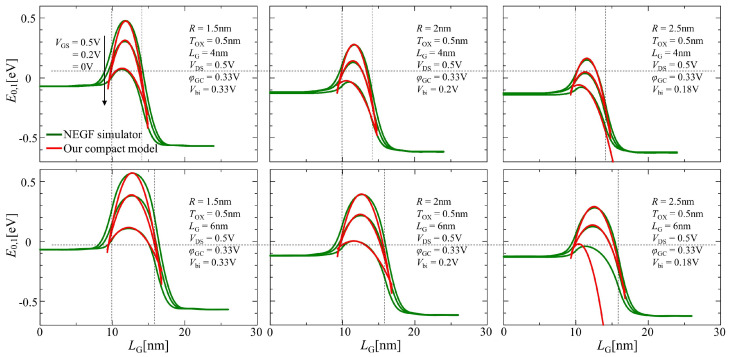
Comparison of the results of the lowest sub-band profile plot between the compact model using Equation (Equation 13) and NEGF simulation under different gate bias conditions.

**Figure 5 nanomaterials-12-03401-f005:**
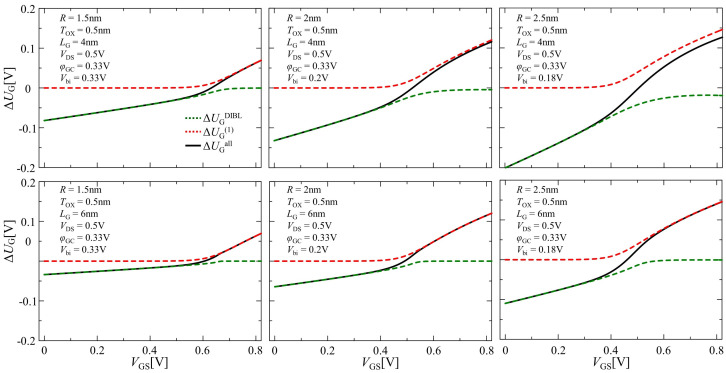
Comparison of the results of $\Delta U_{G}^{\mathrm{all}}$, $\Delta U_{G}^{\mathrm{DIBL}}$ and $\Delta U_{G}^{\left(1\right)}$.

**Figure 6 nanomaterials-12-03401-f006:**
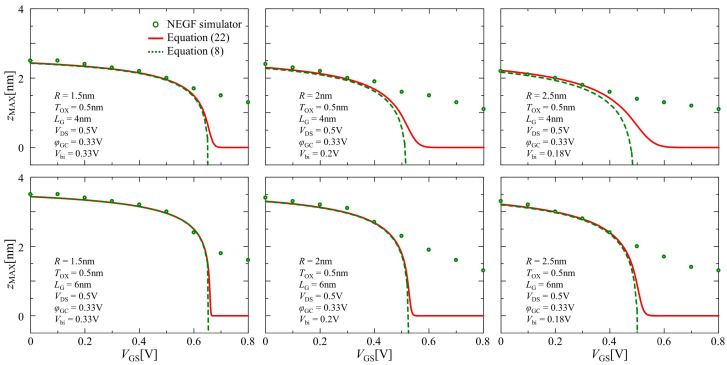
Gate bias dependence of the potential barrier top position along the channel calculated with the NEGF simulation (open circle) and our compact model using Equations (Equation 8) (dashed line) and (Equation 23) (solid curve).

**Figure 7 nanomaterials-12-03401-f007:**
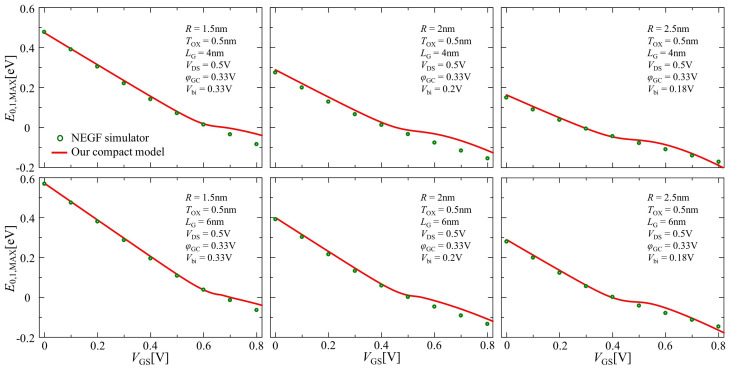
Comparison of the results of the lowest sub-band energy level at the barrier top between NEGF simulation (open circle) and our compact model (solid line) for the lowest sub-band.

**Figure 8 nanomaterials-12-03401-f008:**
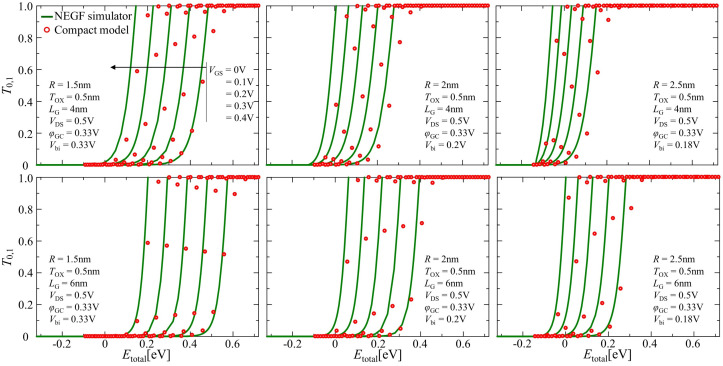
Comparison of the transmission coefficient calculations results between the NEGF simulation (solid line) and our numerical compact model (open circle) for the lowest sub-band.

**Figure 9 nanomaterials-12-03401-f009:**
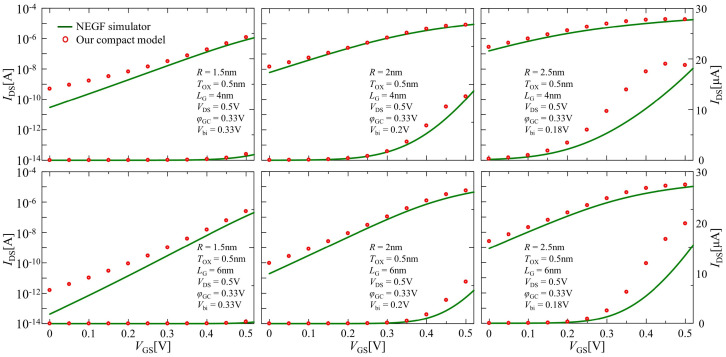
Comparison results between the compact model (open circle) considering the Landauer formula and NEGF simulation (solid line) for drain current characteristic calculations in all operating regions.

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
