# Peer review of "A New Approach to Modeling Ultrashort Channel Ballistic Nanowire GAA MOSFETs"

_nanomaterials, 2022, doi:10.3390/nano12193401_

Round 1
Reviewer 1 Report
This paper demonstrates their previous study regarding the compact analytic model for the subthreshold region for DIBL and source-to-drain tunneling effects. The Gate-All-Around (GAA) transistor is a unique model for future electronics introduced by the pioneer chip makers such as Intel, Samsung, and TSMC. As a diameter is smaller and faces Moore’s law’s limit, GAA is a model that may open up a new pathway for future electronics development. Therefore, the reviewer suggests that this work could be accepted after some minor revision.
1) A paper should be able to readable regardless of the audience’s background. Please kindly put a full abbreviation of GAA, MOSFET, and DIBL in the introduction. They are an obvious abbreviation for the people in this field, but most readers are not.
2) Authors may introduce a motivation for using this model for GAA. There is a short introduction to the DIBL model, but there are many alternative numerical models than DIBL. Please kindly explain why you chose the DIBL model for this analysis. There is some explanation for why the authors chose the DIBL model in the line from 32 to 46, but the reviewer believes this is not enough to convince the robustness of this model.
3) As GAA is a great competitor of Fin-FET, reviewers wonder if authors can add more information and comparison data/words to the introduction to be more attractive to readers.
4) Can authors update some references in the references to the most up-to-date ones? Some references from early 2000 may not be enough to support the novelty of this work.
Reviewer 2 Report
This manuscript reports an extensive theoretical study of drain current in the ballistic mode in the short nanowire-based MOSFETs. The WKB approximation is used to evaluate transmission coefficients for the source-to drain tunneling transitions. The drain current is calculated depending on the relevant parameters of the system. The model results are checked against numerical simulations. The results are relevant and important for the design pf the nanowire-based MOSFETs. The paper is well-written overall. However, the manuscript my benefit from a minor language check. The paper can be published after a minor revision which should address the following points:
11) The acronyms such as “DIBL” in the abstract should be introduced in the main text.
22) The parabolic function describing the potential distribution of the potential along the ultra-short channel should be discussed in the test in more detail. It is not enough to say that it’s introduced “according to our previous study” (page 5).
